# Adverse Perinatal Outcomes among Adolescent Pregnant Women Living with HIV: A Propensity-Score-Matched Study

**DOI:** 10.3390/ijerph20085447

**Published:** 2023-04-10

**Authors:** Gilmar de Souza Osmundo Junior, Fábio Roberto Cabar, Stela Verzinhasse Peres, Adriana Lippi Waissman, Marco Aurélio Knippel Galletta, Rossana Pulcineli Vieira Francisco

**Affiliations:** 1Disciplina de Obstetricia, Departamento de Obstetricia e Ginecologia, Faculdade de Medicina FMUSP, Universidade de Sao Paulo, Sao Paulo 05403-000, Brazil; 2Divisao de Clinica Obstetrica, Hospital das Clinicas HCMFUSP, Faculdade de Medicina, Universidade de Sao Paulo, Sao Paulo 05403-000, Brazil

**Keywords:** HIV, low birth weight, perinatology, pregnancy in adolescence, pregnancy outcome, premature birth

## Abstract

HIV infection and adolescent pregnancy are known to increase the risk of adverse perinatal outcomes. However, data are limited concerning the outcomes of pregnancies among adolescent girls living with HIV. This retrospective propensity-score matched study aimed to compare adverse perinatal outcomes in adolescent pregnant women living with HIV (APW-HIV-positive) with HIV-negative adolescent pregnant women (APW-HIV-negative) and adult pregnant women with HIV (PW-HIV). APW-HIV-positive were propensity-score matched with APW-HIV-negative and PW-HIV. The primary endpoint was a composite endpoint of adverse perinatal outcomes, comprising preterm birth and low birth weight. There were 15 APW-HIV-positive and 45 women in each control group. The APW-HIV-positive were aged 16 (13–17) years and had had HIV for 15.5 (4–17) years, with 86.7% having perinatally acquired HIV. The APW-HIV-positive had higher rates of perinatally acquired HIV infection (86.7 vs. 24.4%, *p* < 0.001), a longer HIV infection time (*p* = 0.021), and longer exposure to antiretroviral therapy (*p* = 0.034) compared with the PW-HIV controls. The APW-HIV-positive had an almost five-fold increased risk of adverse perinatal outcomes compared with healthy controls (42.9% vs. 13.3%, *p* = 0.026; OR 4.9, 95% CI 1.2–19.1). The APW-HIV-positive and APW-HIV-negative groups had similar perinatal outcomes.

## 1. Introduction

Adolescent pregnancy is a global public health issue. The United Nations sexual and reproductive health agency estimates an annual incidence of 16 million adolescent pregnancies [1,2]. In Brazil, the rate of adolescent pregnancies is estimated to be one million per year, corresponding to 17.5% of all deliveries [3]. The Joint United Nations Programme on HIV/AIDS (UNAIDS) also estimates that there are one million adolescent girls living with HIV worldwide [4]. The incidence rate of HIV infection among young Brazilian women is 18.6 per 100,000 [5].

Adolescent pregnancy is defined as pregnancy occurring in girls younger than 19 years old [2], and it is associated with an increased risk of poor social support, delayed access to prenatal care, emotional stress, and poor nutrition [6]. Previous reports have also highlighted an increased prevalence of psychological suffering, physical and sexual violence, and unsafe home environments among pregnant teenagers [6,7].

Previous studies have shown an increased rate of adverse maternal and neonatal outcomes among adolescent pregnant girls [2,8,9,10]. A multicountry study by the World Health Organization (WHO) concluded that adolescent pregnancy is associated with higher risks of pre-eclampsia and eclampsia, preterm birth, and low birth weight [1]. It is also known that adolescent pregnancy is related to an increased incidence of maternal mortality, postpartum hemorrhage, gestational anemia, cephalopelvic disproportion, and postpartum depression [11,12].

Furthermore, HIV infection is also associated with adverse maternal and neonatal outcomes [10,13]. A recent North American retrospective cohort study found an increased probability of preterm premature rupture of membranes, postpartum sepsis, venous thromboembolism, preterm birth, and low birth weight among adult pregnant women living with HIV (PW-HIV) [14]. Tukei et al. [15] found an approximate 21% rate of adverse perinatal events among pregnant women living with HIV, such as stillbirth, preterm delivery, and small-for-gestational-age newborns.

Nonetheless, there are few studies on the perinatal outcomes in adolescent pregnant women living with HIV (APW-HIV-positive) [10,16,17]. Adolescent girls living with HIV have higher rates of unplanned pregnancies than adult women living with HIV [18,19]. Moreover, adolescent girls living with HIV present lower rates of adherence to dual protection for the prevention of pregnancy and HIV transmission [18]. Previous studies have found rates of planned pregnancy less than 15% among APW-HIV-positive [18,19].

Cross-sectional studies suggest high rates of preterm birth and low birth weight in APW-HIV-positive [16,17,20]. On the other hand, the results from a recent North American urban cohort of women living with HIV suggest lower rates of preterm birth among young pregnant women living with HIV when compared with adult PW-HIV [21].

Both adolescent pregnancy and HIV infection during adolescence are common global public health issues, and information on APW-HIV-positive remains limited. This study aimed to compare APW-HIV-positive perinatal outcomes with those of adolescent pregnant women who were HIV negative (APW-HIV-negative) and adult pregnant women living with HIV (PW-HIV) to help better understand the risks for APW-HIV-positive. This is the first Latin American study specifically addressing adolescent pregnancy in women living with HIV.

## 2. Materials and Methods

In this single-center propensity-score-matched observational study, we performed a database search to identify all APW-HIV-positive who had been followed in our Obstetrics Department between 2006 and 2019. Two control groups were selected for comparisons, namely, APW-HIV-negative and PW-HIV groups. Adolescent pregnancy is defined by the WHO as a pregnancy occurring in a girl younger than 19 years old [2]. However, only pregnant girls younger than 18 years old are followed at our high-risk pregnancy out clinic. Thus, in this study, the term adolescent pregnant women comprises girls younger than 18 years old.

A database search was also employed to identify all PW-HIV and APW-HIV-negative patients who had been evaluated at our department during the study period and that could be eligible to be in the control groups. Cases of multiple pregnancies, lost to follow-up, an absence of prenatal care, non-utilization of antiretroviral therapy (ART) during pregnancy, and unavailability of hospital charts were considered as non-inclusion criteria.

Patient medical records were examined and data regarding maternal demographic characteristics, HIV infection, ART exposure, and obstetric and neonatal outcomes were obtained and assessed. The HIV-related aspects considered were: type of transmission (perinatally vs. behaviorally), time of diagnosis, opportunistic infections, viral load (VL), and CD4+ cell count. Laboratory analyses were evaluated at two time points, namely, at baseline (first appointment) and at 34 weeks of gestation.

The assessed perinatal outcomes consisted of gestational age at delivery, preterm birth, birth weight, pre-eclampsia, preterm labor, and HIV mother-to-child transmission. To evaluate adverse outcomes, our primary endpoint was a composite endpoint of adverse perinatal outcomes consisting of low birth weight and/or preterm birth. Low birth weight was defined as birth weight <2500 g, irrespective of the gestational age of delivery. Preterm birth was considered as a spontaneous or indicated birth at <36 + 6 gestational weeks. Thus, patients presenting a birth weight <2500 g and/or delivery at a gestational age <36 + 6 weeks would be classified as having an adverse perinatal outcome.

The care of all pregnant women living with HIV followed the current Brazilian guideline for antenatal care and prevention of mother-to-child HIV transmission in the study period. Patient follow-ups consisted of regular prenatal appointments, morphology scans, and regular assessment of fetal growth and wellbeing. All patients were prescribed ART, which consisted of three active antiretroviral drugs. The patients were assisted by a multidisciplinary team consisting of obstetricians, infectologists, psychologists, and social workers.

Historically, local practices regarding pregnant women living with HIV tended toward elective cesarean delivery, irrespective of a patient’s VL. Such policies have changed and vaginal births have recently been encouraged. During the study period, intravenous zidovudine intrapartum prophylaxis and neonatal zidovudine syrup were routinely recommended to all the patients. Breastfeeding was contraindicated.

### 2.1. Propensity-Score Matching

Control selection was based on a propensity-score (PS) strategy to minimize bias. We performed a covariate balancing PS considering variables that would be available in patients’ medical records and that could influence perinatal outcomes, such as maternal age, body mass index (BMI), and year of delivery. Missing data were also considered.

Initially, we estimated the propensity scores from each individual in our population based on a logistic regression model, considering the variables maternal age, BMI, and year of delivery. After estimating the PS values for all of the available population, we ran an algorithm so that APW-HIV-positive patients could be matched to controls with the most similar PS values (nearest neighbor algorithm) [22,23,24]. Due to the uneven number of APW-HIV positive, APW-HIV-negative, and PWHIV patients, a PS matching at a ratio of 1:3 was performed for each of the control groups.

### 2.2. Statistical Analysis

Categorical data were presented as absolute number and frequencies. Variables presenting normal distributions were expressed as mean and standard deviation; variables with non-normal distributions were expressed as median and interquartile range. Fisher’s exact and chi-squared tests were applied for comparisons between categorical variables. Student’s t and Mann–Whitney U tests were employed for comparisons between continuous variables. Unconditional logistic regression was applied to estimate odds ratios (ORs) and 95% confidence intervals (CIs). A two-tailed *p*-value < 0.05 was considered significant. Data were analyzed using SPSS Statistics for Windows, Version 20.0 (IBM Corp., Armonk, NY, USA) software, and STATA for Windows, Version 13.0 (College Station, TX, USA) software was employed for propensity-score matching.

Manuscript elaboration is in accord with the STROBE Statement [25].

## 3. Results

A total of 15 APW-HIV-positive were identified during the study period. There were also 253 individuals in the APW-HIV-negative group and 226 PW-HIV. All the APW-HIV-positive presented between 2010 and 2014. The average age of the APW-HIV-positive was 16 (range, 13–17) years, they were nulliparous, and they had been living with HIV for 16 (range, 4–17) years. Perinatally acquired HIV was the most common type of HIV transmission among adolescent girls, corresponding to 86.7% (n = 13) of the APW-HIV-positive cases.

The APW-HIV-positive and the APW-HIV-negative had similar socio-demographic characteristics (Table 1), which included low rates of planned pregnancies and stable partnerships. In comparison with the PW-HIV, the APW-HIV-positive were more frequently nulliparous and were less frequently in stable partnerships. Among APW-HIV-positive, only two cases had pre-existing diseases: one case of mild asthma and one case of asymptomatic repaired patent arterial ductus. The APW-HIV-negative group presented three cases of neoplasias in complete remission, one case of asthma, one case of hypothyroidism, and one case of hyperthyroidism. Pre-existing pathologies were more common among PW-HIV, there were: two cases of asthma, two cases of hepatitis C co-infection, two cases of mood disorders, one case of systemic hypertension, one case of hypothyroidism, and one case of chronic kidney disease.

The APW-HIV-positive group presented higher rates of perinatally acquired HIV infection (86.7 vs. 24.4%, *p* < 0.001), a longer HIV infection time (*p* = 0.021), and longer exposure to ART (*p* = 0.034) compared with the PW-HIV controls (Table 2). The APW-HIV-positive and the controls presented with similar CD4 cell count levels at baseline and at 34 weeks (Figure 1). The APW-HIV-positive and the PW-HIV presented undetectable VL at 34 weeks in 50% and 72.5% of cases, respectively (*p* = 0.188). Among those patients with detectable VL, VL levels at 34 weeks were lower in the APW-HIV-positive than in the PW-HIV (*p* = 0.014). There were four cases of opportunistic infection (OI) among the PW-HIV group and none among the APW-HIV-positive.

Regarding perinatal outcomes, there were 14 live births and one miscarriage in the APW-HIV-positive. The average gestational age at delivery was 37.6 (36.3–38.6) weeks, and the average birth weight was 2600 g (2380–3140 g, Table 3). The APW-HIV-negative had higher birth weights (3100 g, range 1520–4318 g, *p* = 0.02); however, they delivered later than the APW-HIV-positive since they were permitted to wait for spontaneous labor, whereas the APW-HIV-positive had planned elective C-sections at between 37 and 38 weeks of gestation.

The rates of low birth weight were 28.6% (n = 4) among APW-HIV-positive, 11.1% (n = 5) among APW-HIV-negative, and 25% (n = 11) in the group consisting of PW-HIV. Regarding preterm birth, there were three cases (21.4%) among APW-HIV-positive, three cases among APW-HIV-negative (6.7%), and 10 cases (22.7%) in the group consisting of PW-HIV. 

The composite endpoint of adverse perinatal outcomes was more frequent among the APW-HIV-positive compared with the APW-HIV-negative (42.9% vs. 13.3, OR 4.9 [95% CI 1.2–19.1%]; *p* = 0.026). The APW-HIV-positive and the PW-HIV had similar perinatal outcomes, including birth weight, gestational age at delivery, and in relation to the composite endpoint of adverse perinatal outcomes (Table 3).

Both the APW-HIV-positive and the APW-HIV-negative groups had similar rates of pre-eclampsia (*p* = 1.0) and threatened preterm labor (*p* = 0.564). There were no cases of mother-to-child HIV transmission.

## 4. Discussion

This single-center, propensity-score-matched observational study involved two retrospective cohorts of patients who had been followed at a university hospital in Brazil between 2006 and 2019: women living with HIV and adolescent pregnant women. Adverse maternal and neonatal outcomes from APW-HIV-positive were evaluated in comparison with two groups of women, namely, APW-HIV-negative and PW-HIV. All patients were followed in the same prenatal service during the study period. In this study, the APW-HIV-positive presented with an almost five-fold increased risk of the composite adverse perinatal outcome than the APW-HIV-negative.

Previous studies have hypothesized that the higher incidence of adverse obstetric outcomes among adolescents is linked to socioeconomic factors and to the vulnerability of such a population [26,27]. Nonetheless, according to our findings, the APW-HIV-positive and the APW-HIV-negative had similar socio-demographic characteristics, which included low rates of planned pregnancies and stable partnerships, yet HIV infection remained the most important risk factor for adverse outcomes.

It has previously been well established that pregnancy in healthy adolescence presents higher maternal and neonatal risks when compared with pregnancy during adult ages. Zhang et al. [28] showed that adolescents have a higher risk of preterm delivery, neonates small for gestational age, stillbirth, and neonatal death compared with adult women in Asia. Similarly, low birth weight and preterm birth rates were reported to be higher among adolescents than adult women in an Ethiopian study [29]. Nevertheless, in our study, the APW-HIV-positive and the PW-HIV had similar frequencies of obstetric and perinatal complications.

Classically, it is well known that women living with HIV who are not under ART have higher risk of spontaneous and indicated preterm birth, small-for-gestational-age newborns, very-small-for-gestational-age newborns, stillbirth, and neonatal death [9,30,31]. Although previous studies have shown that ART reduces the risks of adverse perinatal outcomes [32,33], those patients receiving adequate ART are still at a 2.2-fold increased risk of spontaneous preterm delivery and at an almost 1.9-fold increased risk of term low birth weight [9]. Our findings accord with those of previous studies that have shown that newborns of women living with HIV have poorer birth outcomes than those who are uninfected, even when they have viral suppression and high CD4 cell counts, regardless of gestational age at delivery [30].

In our study population, the APW-HIV-positive and the PW-HIV had similar VL and CD4+ cell count levels at baseline and at 34 weeks, which may have contributed to similar rates of adverse perinatal events. Some previous studies considering low birth weight and preterm birth among patients living with HIV have suggested that a high VL and low levels of CD4 are contributing risk factors for adverse perinatal outcomes; however, this remains controversial [34,35,36].

It is also noteworthy that APW-HIV-positive and PW-HIV positive had similar VL and CD4+ cell count levels in our population, since APW-HIV-positive were more antiretroviral experienced, had longer time periods of HIV infection, and had a higher proportion of perinatally acquired HIV in comparison with PW-HIV. Adolescent patients with perinatally acquired HIV are also known to have a higher incidence of poor compliance with antiretroviral drugs [37,38]. Previous studies found lower rates of viral suppression among adolescent patients living with HIV [39,40]. Van Wyk et al. also suggest that adolescents aged 15–19 years have the poorest compliance with HIV treatment [41]. Our population comprises a small sample size of APW-HIV-positive; thus, absence of significant statistical difference between viral suppression rates from both groups might just reflect an underpowered analysis.

Rates of viral suppression at the first prenatal appointment were 21.4% and 40.5% for APW-HIV-positive and PW-HIV, respectively. It is important to highlight that those cases occurred between 2010 and 2014, when ART was limited to patients with AIDS-defining conditions and pregnant women. WHO’s first recommendation on early indication of ART for every person living with HIV was published in 2013 [42]. The National HIV Surveillance System (NHSS) reported that only 57.9% of people living with HIV in the US had viral suppression in 2014 [43]. Similarly to our results, a recent Brazilian study regarding young pregnant women living with HIV found near-to-delivery viral suppression rates between 42 and 56% [44].

In our population, APW-HIV-positive and PW-HIV groups were not comparable in terms of viral transmission. The rates of perinatally acquired HIV infection were 86.7% in the APW-HIV-positive group and 24.4% among PW-HIV controls. However, recent data show a trend towards the reduction of perinatally acquired cases and the increment of sexually transmitted cases even among teenagers. Data from the Centers for Disease Control and Prevention (CDC) [45] demonstrate that perinatal transmission accounted for half of the new HIV cases in girls between 13 and 14 years old in 2020. Meanwhile, more than 80% of the new HIV cases in girls older than 14 years old were linked to sexual transmission [45].

Our findings are consistent with a recent national population-based study, in which adolescent girls living with HIV were reported to have a higher likelihood of small-for-gestational-age newborns compared with HIV-negative teenagers, although the outcomes were comparable with adult women living with HIV [46].

One strength of our study is that we included a well-defined population of adolescent and adult pregnant women, who were followed in a single medical service at a developing country university hospital. Owing to our study design, we were able to evaluate multiple maternal complications and adverse birth outcomes. Further, we proceeded with a propensity-score controlled analysis.

The present study has some weaknesses inherent to its observational and retrospective nature, such as limited available information on medical records. The selection of individuals to constitute control groups is also a potential source of bias. Furthermore, there is also a risk of confounding. Propensity-score matching is a useful tool to control confounding since it allows balancing groups according to multiple variables [22,23,24]. Propensity-score balancing is especially adequate for studies with small sample sizes and low rates of events, which are limitations for the application of the more traditional logistic regression strategies for controlling confounders [47].

We opted for a composite endpoint because it represents a gain in statistical power in the context of our small sample size. We chose low birth weight and preterm delivery to constitute our composite endpoint because they have a reasonably similar clinical relevance.

In fact, there are several ways to assess adverse perinatal outcomes among APW-HIV-positive, for instance low birth weight, preterm delivery, neonatal death, stillbirth, admission to neonatal care unit, and cerebral palsy. Events such as neonatal death and cerebral palsy have a very low incidence [48,49]; thus, a larger sample size would be necessary to properly analyze them. On the other hand, Apgar score, neonatal death, and admission to a neonatal care unit may have a higher risk of bias, as they depend on the neonatal medical assistance. The authors have opted for choosing low birth weight and preterm delivery as our primary outcomes because they are robust outcomes, they have a clear definition, and they are proven to be associated with higher infant mortality and morbidities, such as bronchopulmonary dysplasia, intraventricular hemorrhage, neurodevelopmental impairment, and recurrent hospital admissions [50,51]. Furthermore, preterm delivery is also related to an increased risk of mother-to-child HIV transmission [52,53,54].

The small sample size is another limitation of our study, since it was not possible to evaluate all possible maternal–fetal–neonatal complications due to the small number of cases included. There is also a potential risk of underpowered statistical analysis. Therefore, statistical tests presenting negative results should be cautiously interpreted in face of the risk of type II statistical errors.

Our observational study does not allow determining a causal relationship between APW-HIV-positive and adverse neonatal outcomes. Future multicenter prospective research could be more accurate in studying the relationship between adolescent pregnancy among girls living with HIV and preterm delivery or low birth weight.

Pregnancy among adolescent girls living with HIV also potentially jeopardizes patients’ compliance with the antiretroviral treatment. A recent East African cohort study demonstrated that APW-HIV-positive who were older than 15 years old had approximately twice the chance of becoming lost to follow-up from HIV clinics than male teenagers [55]. On the other hand, a Brazilian historical cohort study found that a younger maternal age was an independent risk factor for being lost to follow-up for HIV-exposed children during the first 18 months of surveillance [56].

Despite the potential risks for perinatal outcomes, the incidence of pregnancy seems to be similar between adolescent girls living with HIV and uninfected teenagers. A North American prospective cohort study found comparable rates of adolescent pregnancy between HIV-positive and uninfected girls [57]. An older age and not using hormonal contraception were risk factors for adolescent pregnancy among HIV-positive girls [57]. Further, Finocchario-Kessler et al. [58] have demonstrated that HIV-positive and uninfected adolescent girls had similar rates of pregnancy desire and positive childbearing motivation among African American girls in the US. 

The increased rates of adverse perinatal outcomes among women living with HIV highlight the importance of accessible pregnancy/family planning and contraceptive methods, routine HIV screening, and accessible HIV prevention methods for women of reproductive age. Moreover, the Brazilian 1982 Birth Cohort Study [59] evaluated the long-term impacts of adolescent pregnancy, finding that women who gave birth during their adolescence had fewer years of schooling and lower incomes at 30 years of age. Thus, adolescent pregnancy in the context of girls living with HIV is potentially associated with negative impacts on maternal socioeconomic outcomes, compliance with HIV treatment, follow-up of the exposed infant and, according to our results, to an increased risk of adverse perinatal outcomes.

Therefore, public health agents and health care providers who work with girls living with HIV should focus on measures to prevent adolescent pregnancy. HIV-infected teenagers should be counseled towards the use of dual protection, consisting of condom usage associated with a highly effective contraceptive method, specially intrauterine devices, progestin implants, and progestin-only injectable contraception [60]. Further, strategies regarding sexual education, girls’ empowerment, and even economic empowerment are pivotal to prevent adolescent pregnancy among girls living with HIV.

## 5. Conclusions

Our findings suggest that APW-HIV-positive were at increased risk of adverse perinatal outcomes when compared with APW-HIV-negative. Nonetheless the risks of perinatal adverse outcomes appear to be comparable with PW-HIV. Therefore, it is advisable to encourage sexual education among this population to prevent unplanned pregnancies with a higher risk of adverse outcomes. It is recommended that future research involves multicenter studies with larger sample sizes, which would increase statistical power and help facilitate detection of less common adverse outcomes and their risk factors.

## Figures and Tables

**Figure 1 ijerph-20-05447-f001:**
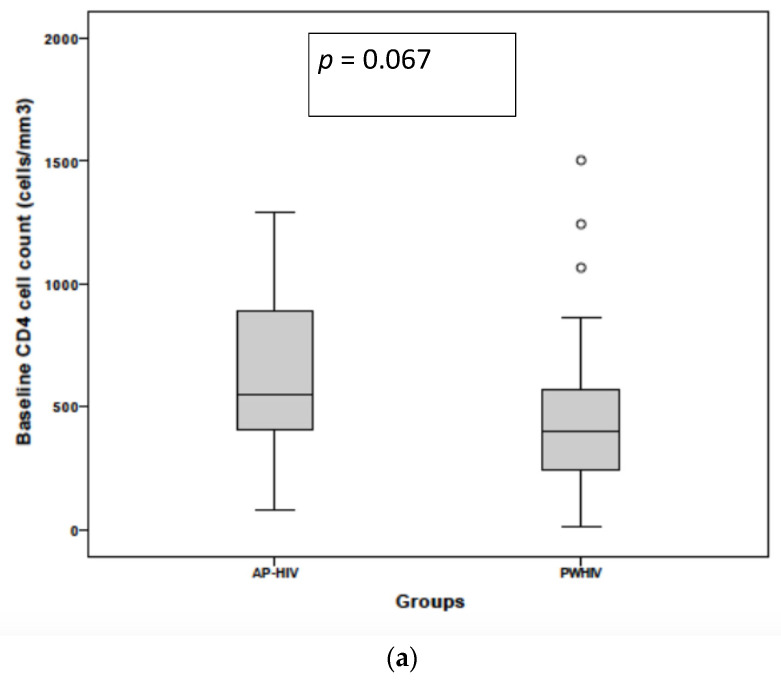
(**a**) A comparison between baseline CD4 cell counts among adolescent pregnant women living with HIV (AP-HIV) and adult women living with HIV (PW-HIV). (**b**) A comparison between CD4 cell counts at 34 weeks of gestation among adolescent pregnant women living with HIV (AP-HIV) and adult women living with HIV (PW-HIV).

**Table 1 ijerph-20-05447-t001:** Socio-demographic characteristics of cases and controls (n = 105).

	APW-HIV-Positive (n = 15)	APW-HIV-Negative (n = 45)	*p* ^1^	PW-HIV (n = 45)	*p* ^2^
Variables	Cases/Total Available (%)	Cases/Total Available (%)		Cases/Total Available (%)	
Age, years, median (min–max)	16 (13–17)	16 (13–17)	0.720	29 (19–42)	<0.001
BMI, kg/m^2^, average (±SD)	23.1 (±2.3)	22.8 (±2.1)	0.691	22.5 (±2.2)	0.420
Nulliparous, n (%)	15/15 (100%)	43/45 (95.6%)	1.0	23/45 (51.1%)	0.001
Stable partnership, n (%)	3/14 (21.4%)	12/45 (26.7%)	1.0	28/45 (62.2%)	0.008
Planned pregnancy, n (%)	1/14 (7.1%)	8/45 (17.8%)	0.671	6/43 (14%)	0.669
Smoking, n (%)	2/14 (14.3%)	9/45 (20%)	1.0	13/44 (29.5%)	0.317
Drugs, n (%)	2/14 (14.3%)	1/42 (2.4%)	0.151	8/44 (18.2%)	1.0
GA at first appointment, weeks, average (± SD)	17.4 (±7.5)	17.9 (±6.8)	0.827	15.8 (±7.2)	0.669
Syphilis, n (%)	0/15	0/44	-	3/45 (6.7%)	0.566

APW-HIV-positive, adolescent pregnant women with HIV; APW-HIV-negative, adolescent HIV-negative pregnant women; BMI, body mass index; GA, gestational age; PW-HIV, adult pregnant women living with HIV; SD, standard deviation; ^1^ *p*-Value resulting from a comparison between APW-HIV-positive and APW groups; ^2^ *p*-Value resulting from a comparison between APW-HIV-positive and PW-HIV groups.

**Table 2 ijerph-20-05447-t002:** HIV-infection-related features of adolescent and adult pregnant women living with HIV (n = 60).

	APW-HIV-Positive (n = 15)	PW-HIV (n = 45)	*p*
Variables	Cases/Total Available (%)	Cases/Total Available (%)	
Time of HIV infection, years, median (min–max)	16.0 (4–17)	10 (0–22)	0.021
Perinatally acquired HIV infection, n (%)	13/15 (86.7%)	11/45 (24.4%)	<0.001
Time of ART, years, median (min–max)	13.5 (0–16)	3.5 (0–21)	0.034
Previous opportunistic infection, n (%)	2/13 (15.4%)	13/45 (28.9%)	0.480
Opportunistic infection in pregnancy, n (%)	0/15	4/45 (8.9%)	0.564
Baseline CD4+ cell count, /mm (3), median (min–max)	546 (80–1288)	399 (13–1503)	0.067
Undetectable baseline VL, n (%)	3/14 (21.4%)	17/42 (40.5%)	0.198
VL levels baseline, log, mean (±SD) ^1^	2.9 (±1.8)	3.7 (±0.9)	0.150
CD4+ cell count at 34 weeks, /mm (3), median (min–max)	512.5 (160–1103)	542 (17–1280)	0.686
Undetectable VL at 34 weeks, n (%)	7/14 (50%)	29/40 (72.5%)	0.188
VL levels at 34 weeks, log, mean (± SD) ^1^	1.5 (±1.8)	3.2 (±1.3)	0.014

APW-HIV-positive, adolescent pregnant women living with human immunodeficiency virus; ART, antiretroviral therapy; PW-HIV, adult pregnant women living with HIV; SD, standard deviation; VL, viral load; ^1^ among patients with detectable viral loads.

**Table 3 ijerph-20-05447-t003:** Perinatal outcomes among cases and controls (n = 105).

	APW-HIV-Positive (n = 15)	APW-HIV-Negative (n = 45)	*p* ^1^	PW-HIV (n = 45)	*p* ^2^
Variables	Cases/Total Available (%)	Cases/Total Available (%)		Cases/Total Available (%)	
GA at delivery, weeks, median (min–max)	37.6 (36.3–38.6)	39.0 (32.2–40.3)	0.001	37.6 (22.1–40)	0.955
Birth weight, gram, median (min–max)	2600 (2380–3140)	3100 (1520–4318)	0.002	2900 (500–3750)	0.201
Vaginal delivery, n (%)	0/14	30/45 (66.7%)	<0.001	2/43 (4.7%)	1.0
Pre-eclampsia, n (%)	0/15	3/45 (6.7%)	1.0	1/44 (2.3%)	1.0
Preterm labor, n (%)	1/14 (7.1%)	2/45 (4.4%)	0.564	5/44 (11.4%)	1.0
Composite endpoint of adverse perinatal outcomes, n (%)	6/14 (42.9%)	6/45 (13.3%)	0.026	13/44 (29.5%)	0.514

APW-HIV-positive, adolescent pregnant women living with HIV; APW-HIV-negative, adolescent HIV-negative pregnant women; GA, gestational age; LBW, low birth weight; PW-HIV, adult pregnant women living with HIV; ^1^ *p*-value resulting from a comparison between APW-HIV-positive and APW-HIV-negative; ^2^ *p*-value resulting from a comparison between APW-HIV-positive and PW-HIV groups.

## Data Availability

The data supporting the findings of this study are available from the corresponding author (GSOJ) upon reasonable request.

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
