# Peer review of "Adverse Perinatal Outcomes among Adolescent Pregnant Women Living with HIV: A Propensity-Score-Matched Study"

_ijerph, 2023, doi:10.3390/ijerph20085447_

Round 1

Reviewer 1 Report

General:

This topic is interesting that aimed to compare APW-HIV-positive perinatal outcomes with those of adolescent 51 pregnant women who were HIV-negative (APW-HIV-negative) and adult pregnant 52 women with HIV (PW-HIV) to help better understand the risks for APW-HIV-positive.

In detail:

Materials and Methods

  1. Please add the citation of the statement (line 87): The Nearest Neighbor algorithm was employed for propensity score matching.
  2. What is the age criteria for adolescent female? Please state. Because adolescents younger than 16 years have been shown to be at an increased risk of adverse pregnancy outcomes.
  3. Potential confounding factors, such as the time of pregnancy shortly after menarche (<2 years after menarche, defined as low gynecological age) are at an increased risk of adverse outcomes. Therefore, maternal age could be categorized into several groups.
  4. Medical conditions during pregnancy (such as anemia, thalassemia, malaria, heart problem, or other co-morbidities) also may have a role as confounding factors. If you have data, it will be better
  5. Do you have data on the educational status (educational attainment) of the respondent?
  6. Cases of multiple pregnancies, why excluded?
  7. Please explain more about adverse perinatal events. Is it also include severe neonatal conditions such as asphyxia or early neonatal death? Data about the Apgar score is also crucial as the baby's outcome.
  8. Cephalo-pelvic disproportion is also one of the adverse maternal and perinatal outcomes. Authors may consider this in their analysis.

Author Response

We thank your and the editor`s comments. We have replied all the comments and made the appropriate suggested modifications, which are highlighted in the text.

We have also edited the manuscript to fulfill the requirement of a minimum of 4000 words. New text inserts are highlighted as well.

We hope you will find our paper suitable for publication in the International Journal of Environmental Research and Public Health.

With best wishes,

Reviewer 1

  1. Please add the citation of the statement (line 87): The Nearest Neighbor algorithm was employed for propensity score matching.

We agree that we should present a more detailed explanation about the propensity-score strategy. Although propensity-score is useful and valid for controlling covariates and reduce bias, it is not as commonly employed as logistic regression. Thus, we have re-written the propensity-score matching section, providing more detailed explanations and including references: Controls’ selection was based on a propensity-score (PS) strategy to minimize bias. We performed a covariate balancing PS considering variables that would be available at patients’ medical records and that could influence the perinatal outcomes, such as maternal age, body mass index (BMI), and year of delivery. Missing data were also considered. After estimating the PS values for all the available population, we ran an algorithm that cases could be matched to controls with the most similar PS values (nearest neighbor algorithm).”

  1. What is the age criteria for adolescent female? Please state. Because adolescents younger than 16 years have been shown to be at an increased risk of adverse pregnancy outcomes.

In fact, the definition of adolescent pregnancy was missing. We have now included the World Health Organization’s definition of adolescence and we also added a statement explaining that we have only included patients younger than 18 years old in our study.

Adolescent pregnancy is defined by the WHO as a pregnancy occurring in a girl younger than 19 years old.1 However, only pregnant girls < 18 years old are followed at our high-risk pregnancy outclinic. Thus, in this study, the term adolescent pregnant women comprises girls younger than 18 years old.”

  1. Potential confounding factors, such as the time of pregnancy shortly after menarche (<2 years after menarche, defined as low gynecological age) are at an increased risk of adverse outcomes. Therefore, maternal age could be categorized into several groups

We completely agree that confounding factors could influence the outcomes and that we should control these factors for analyses. Propensity-score is a useful tool for controlling confounding factors when we have a large number of covariates and/or a small number of events (as in our case) – which would be a limiting factor for logistic regression. Further, the propensity-score has the advantage of considering continuous variables, so we do not need to artificially categorize them into strata. Thus, in our propensity-score matching strategy, we are already controlling for maternal age.

We agree that pregnancy shortly after menarche could be a confounding factor. However, as a limitation of retrospective studies, this information is not available in our medical records.

  1. Medical conditions during pregnancy (such as anemia, thalassemia, malaria, heart problem, or other co-morbidities) also may have a role as confounding factors. If you have data, it will be better

We agree that medical conditions could interfere with perinatal outcomes. However, only two APW-HIV-positive cases had preexisting diseases: one case of asthma and one case of asymptomatic repaired patent arterial ductus (both cases were mild conditions). However, we have included a more detailed explanation about cases’ and controls’ preexisting diseases.

“Among APW-HIV-positive, only two cases had preexisting diseases: one case of mild asthma and one case of asymptomatic repaired patent arterial ductus. APW-HIV–negative group presented three cases of neoplasias in complete remission, one case of asthma, one case of hypothyroidism, and one case of hyperthyroidism. Preexisting pathologies were more common among PW-HIV: two cases of asthma, two cases of hepatitis C co-infection, two cases of mood disorders, one case of systemic hypertension, one case of asthma, one case of hypothyroidism, and one case of chronic kidney disease.”

  1. Do you have data on the educational status (educational attainment) of the respondent?

Unfortunately, we do not have data on education status.

  1. Cases of multiple pregnancies, why excluded?

Twin pregnancies are known to have an increased risk of adverse perinatal outcomes, such as preterm delivery, low birth weight, stillbirth, neonatal ICU admission. Thus, it would not be adequate to compare outcomes from singleton and multiple pregnancies.

To proceed with the analysis of perinatal outcomes among twin pregnancies of HIV-positive pregnant women, we would need a very larger sample size – which is not feasible in such a specific population (pregnant adolescent women living with HIV).

  1. Please explain more about adverse perinatal events. Is it also include severe neonatal conditions such as asphyxia or early neonatal death? Data about the Apgar score is also crucial as the baby's outcome

In fact, there are several ways to assess an adverse perinatal outcome: low birth weight, preterm delivery, neonatal death, stillbirth, admission to neonatal care unit, cerebral palsy, etc. Events such as neonatal death and cerebral palsy have a very low incidence; thus, it would be necessary a larger sample size to properly analyze them. On the other hand, Apgar score, neonatal death, and admission to neonatal care unit may have a higher risk of bias, as they depend on the neonatal medical assistance.

We also know that the larger the number of analyzed variables the higher the probability of false-positive results.

We have opted for choosing low birth weight and preterm delivery as our primary outcomes because they are robust outcomes, they have a clear definition, and they are proven to be associated with higher infant mortality and short- and long-term morbidities, such as bronchopulmonary dysplasia, intraventricular hemorrhage, chronic diseases, neurodevelopmental impairment, and recurrent hospital admissions. Furthermore, these outcomes are classically adopted among previous published papers regarding pregnant women living with HIV.

  1. Cephalo-pelvic disproportion is also one of the adverse maternal and perinatal outcomes. Authors may consider this in their analysis.

We agree that cephalopelvic disproportion (CPD) is a relevant adverse maternal outcome, as well as emergency cesarean section. However, our rate of vaginal delivery is < 5%. We would not have an adequate statistical power to evaluate outcomes related to vaginal delivery, such as CPD.

It is important to highlight that we currently encourage vaginal delivery among HIV-positive patients with low values of viral load. Nonetheless, Local practices regarding pregnant women living with HIV tended toward elective cesarean delivery irrespectively of the VL for many years.

Reviewer 2 Report

The present study aimed to compare prenatal outcomes between 3 groups of pregnant women:

1.       Adolescents with HIV

2.       Adolescents without HIV

3.       Adult women with HIV

The authors assert that this is a case control study.  It is not.  Case-control studies are based on outcomes.  Here the study is based on exposures, namely HIV and age.  Thus, it is a cohort study.

The selection of “control” subjects was based on propensity score matching.  It is not stated what propensity is being considered and how the scores were created.  Typically, this is propensity for exposure generated from multivariable logistic regression.  Matching or weighting would lead to balanced exposure arms.   “A covariate balancing propensity score was estimated considering potential confounders that could interfere with perinatal results, such as age, body mass index, and year of delivery.”  Since age is one of the exposures of interest it does not make sense to try to balance age.

In comparing groups 1 and 3 it would be appropriate to match by HIV transmission and parity.  The adult women should have been restricted to a younger age range say 20-30.  Including much older women (age 42) runs the risk of obscuring findings.

This study is too flawed to be worth publishing.

Author Response

We thank your and the editor`s comments. We have replied all the comments and made the appropriate suggested modifications, which are highlighted in the text.

We have also edited the manuscript to fulfill the requirement of a minimum of 4000 words. New text inserts are highlighted as well.

We hope you will find our paper suitable for publication in the International Journal of Environmental Research and Public Health.

With best wishes,

Reviewer 2

  1. The authors assert that this is a case control study.It is not.Case-control studies are based on outcomes.  Here the study is based on exposures, namely HIV and age.  Thus, it is a cohort study.

We completely agree that it was a misuse of the term “case-control study”. We have now changed the manuscript title and description, employing the term “propensity-score matched observational study”

  1. The selection of “control” subjects was based on propensity score matching.It is not stated what propensity is being considered and how the scores were created.Typically, this is propensity for exposure generated from multivariable logistic regression.  Matching or weighting would lead to balanced exposure arms.   “A covariate balancing propensity score was estimated considering potential confounders that could interfere with perinatal results, such as age, body mass index, and year of delivery.”  Since age is one of the exposures of interest it does not make sense to try to balance age.

Covariate balancing propensity score consists in a mathematical strategy in which several variables are taken into account and each individual would receive a propensity score value based on these variables. We have run an algorithm to match cases to those controls presenting the most similar values of propensity score (nearest neighbor algorithm). We agree that we could have made it more clear in the methods description, so we have changed it: Controls’ selection was based on a propensity-score (PS) strategy to minimize bias. We performed a covariate balancing PS considering variables that would be available at patients’ medical records and that could influence the perinatal outcomes, such as maternal age, body mass index (BMI), and year of delivery. Missing data were also considered. After estimating the PS values for all the available population, we ran an algorithm that cases could be matched to controls with the most similar PS values (nearest neighbor algorithm).”

However, we must slightly disagree according to age balancing. In our case, the exposure of interest is adolescent pregnancy among HIV-infected girls. Age is already a well-known risk factor for adverse perinatal outcome. If we had not balanced age, we could have obtained, for instance, a control group of HIV-negative girls who were older than the HIV-positive patients. And in this scenario, age would result in confounding. That is the reason we opted for including age in the propensity-score balancing.

  1. In comparing groups 1 and 3 it would be appropriate to match by HIV transmission and parity.The adult women should have been restricted to a younger age range say 20-30.Including much older women (age 42) runs the risk of obscuring findings.

Groups 1 (adolescent pregnant women living with HIV – APW-HIV-positive) and 3 (adult women living with HIV – PW-HIV) have different epidemiological features. APW-HIV-positive have higher rates of perinatally acquired HIV and many of them are nulliparous. On the other hand, women with behaviorally acquired HIV tend to be older and multiparous.

Matching these groups according to parity and HIV transmission would not be possible even in a larger sample size.

Reviewer 3 Report

·       The chart review was from 2006-2019, please discuss why HIV is only until 2014. Is it because there were no individuals after 2014, if so should the chart review also in the same time? Or it has to be discussed, it appears that adolescent preganciess have significantly reduced, which is a breakthrough and indicates progress of education or case.

·       Line 119- please change achieved . The authors can only say that at baseline the VL was this and at 34 weeks..The study did not do any intervention to change ART patterns

·       What are the adverse perinatal events. It says it is 6/15 and 6/45. So, is it because of numbers? It appears that many of the outcomes including preeclampsia, birth weight etc. are not different?

·       Are LBW and premature birth part of the adverse events, if so the numbers do not match? please reconcile. Also, if the adverse events include LBW and preterm then the numbers dont match with adverse events. It should be 7/14 and 8/45

·       The reviewer is not able to comment on propensity score matched.

Author Response

We thank your and the editor`s comments. We have replied all the comments and made the appropriate suggested modifications, which are highlighted in the text.

We have also edited the manuscript to fulfill the requirement of a minimum of 4000 words. New text inserts are highlighted as well.

We hope you will find our paper suitable for publication in the International Journal of Environmental Research and Public Health.

With best wishes,

Reviewer 3

  1. The chart review was from 2006-2019, please discuss why HIV is only until 2014. Is it because there were no individuals after 2014, if so should the chart review also in the same time? Or it has to be discussed, it appears that adolescent preganciess have significantly reduced, which is a breakthrough and indicates progress of education or case.

We searched our database for adolescent pregnant women living with HIV (APW-HIV-positive) between 2006 and 2019. However, we did not have any case after 2014.

Several factors could explain the absence of APW-HIV-positive cases after 2014:

  • 87% of our APW-HIV-positive had perinatally acquired HIV. There have been several public health strategies to reduce mother-to-child transmission in the last decades. So, we could hypothesize that a decrease in the number of teenagers living with perinatally-acquired HIV (and adolescent pregnancy);
  • Public policies tending toward decentralization of the public health. Thus, less severe cases of APW-HIV-positive would not be referred to tertiary hospitals (such our service);
  • Our sample comprises only a very small number of patients from one single center. Thus, this observed reduction of cases after 2014 may have occurred just by chance. It would be necessary larger and multicenter studies to an accurate analysis of the populational trends in the incidence of adolescent pregnancy among girls living with HIV.

  1. Line 119- please change achieved . The authors can only say that at baseline the VL was this and at 34 weeks..The study did not do any intervention to change ART patterns

We agree that the word “achieved” is not the most appropriate, since this is an observational study. We have now re-written the sentence: “The APW-HIV-positive and the PW-HIV presented undetectable VL at 34 weeks in 50% and 72.5% of cases...”

  1. What are the adverse perinatal events. It says it is 6/15 and 6/45. So, is it because of numbers? It appears that many of the outcomes including preeclampsia, birth weight etc. are not different?

We defined our primary outcome as a “composite endpoint of perinatal adverse outcomes” – which consists of low birth weight and/or preterm birth. Thus, any patient presenting low birth weight or preterm delivery will be classified as an adverse neonatal event.

We have opted for choosing low birth weight and preterm delivery as our primary outcomes because they are robust outcomes, they have a clear definition, and they are proven to be associated with higher infant mortality and short- and long-term morbidities, such as bronchopulmonary dysplasia, intraventricular hemorrhage, chronic diseases, neurodevelopmental impairment, and recurrent hospital admissions. Furthermore, these outcomes are classically adopted among previous published papers regarding pregnant women living with HIV.

In table 3, we intended to demonstrate that 6/14 (42.9%) cases of APW-HIV-positive had an adverse neonatal event, compared to 6 / 45 (13.3%) of APW-HIV-negative controls.

We agree that Table 3 was not clear enough about the outcome. So, we have now changed it: we present only the number of the composite outcome in each group and we label it as “composite endpoint of adverse perinatal outcomes”.

  1. Are LBW and premature birth part of the adverse events, if so the numbers do not match? please reconcile. Also, if the adverse events include LBW and preterm then the numbers dont match with adverse events. It should be 7/14 and 8/45

In fact, the composite outcome consists of any patient presenting low birth weight and/or preterm delivery. Different scenarios would be classified as “ composite adverse neonatal event”: 1) a patient presenting preterm delivery and a newborn with an adequate birth weight; 2) a patient presenting a low birth weight newborn at term; and 3) a patient presenting preterm delivery and low birth weight. This is the reason why the number of events seem not to match.

We agree that displaying these data in Table 3 without further explanation can lead to misunderstanding. So, we have deleted this information from table 3 and we have included a more detailed explanation in the results section.

Round 2

Reviewer 1 Report

I am satisfied with the author’s responses to my concerns raised in my initial review. I recommend that the revised  paper be accepted.

Reviewer 2 Report

A propensity score with 3 variables is ridiculous.  We are still not told what was predicted in the propensity score model.  A proper analysis is the only thing that could save a study with only 15 "exposed" .  This study is a waste of time.

Reject.

Reviewer 3 Report

The comments are adequetely addressed.